# UniAudio 1.5: Large Language Model-Driven Audio Codec is A Few-Shot Audio Task Learner

**Dongchao Yang[1], Haohan Guo[1], Yuanyuan Wang[2], Rongjie Huang[1], Xiang Li[2]**
**Xu Tan[3], Xixin Wu[1], Helen Meng[1]**
[1] The Chinese University of Hong Kong, [2] Tsinghua University, [3] Microsoft Research Asia
`{dcyang,hmmeng}@se.cuhk.edu.hk`

## Abstract

Large Language models (LLMs) have demonstrated supreme capabilities in textual understanding and generation, but cannot be directly applied to cross-modal tasks without fine-tuning. This paper proposes a cross-modal in-context learning approach, empowering the frozen LLMs to achieve multiple audio tasks in a few-shot style without any parameter update. Specifically, we propose a novel LLM-driven audio codec model, LLM-Codec, which transfers the audio modality into textual space by representing audio tokens with words or sub-words from the LLM vocabulary, while maintaining high audio reconstruction quality. The key idea is to reduce the modality heterogeneity between text and audio by compressing the audio modality into the well-trained textual space of LLMs. Thus, the audio representation can be viewed as a new *foreign language*, and LLMs can learn the new *foreign language* with several demonstrations. In experiments, we investigate the performance of the proposed approach across multiple audio understanding and generation tasks, *e.g.* speech emotion classification, audio classification, text-to-speech generation, speech enhancement, etc. Experimental results show that LLMs equipped with the LLM-Codec, named as UniAudio 1.5, prompted by only a few examples, can perform effectively in simple scenarios, validating our cross-modal in-context learning approach. To facilitate research on few-shot audio task learning and multi-modal LLMs, we have open-sourced the LLM-Codec model. [1]

## 1 Introduction

Large language models (LLMs) (*e.g.*, GPT-4 [2], LLAMA [36]) have become increasingly versatile and effective in handling diverse and complex Natural Language Processing (NLP) tasks as they scale in model size and training data. It is worth noting that the in-context learning ability of LLMs can be used to solve unseen tasks, *e.g.*, we can provide instructions along with a few demonstrations, enabling LLMs to learn and solve new tasks. The success of LLMs inspires the development of multi-modal LLMs, naturally leading to the idea of empowering their auditory capabilities to tackle audio-related tasks. There have been notable advancements in extending the capabilities of LLMs to tackle audio understanding tasks by combining the pre-trained audio encoder (*e.g.* Whisper encoder [31]) and LLMs. For instance, models like WavLLM [15], SALMONN [35], and Qwen-audio [8] propose training multi-modal LLMs by integrating a pre-trained audio encoder, a trainable adaptor, and pre-trained LLMs. They try to align the audio and text modalities by updating the adaptor or fine-tuning the LLMs with LORA [14]. However, previous works have limitations: (1) they primarily focus on expanding LLMs to solve specific audio tasks without leveraging in-context learning for unseen audio tasks; (2) they do not support audio generation tasks, which limits their applicability; (3) they require large-scale audio data for aligning audio and text modalities, increasing the burden of model training and data collection.

---

[1] https://github.com/yangdongchao/LLM-Codec

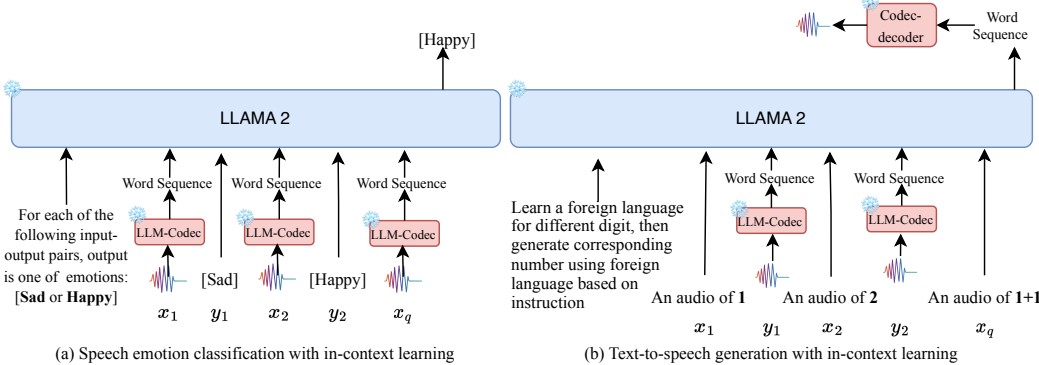

(a) Speech emotion classification with in-context learning   (b) Text-to-speech generation with in-context learning

Figure 1: This figure illustrates the framework of the proposed approach (UniAudio 1.5) for performing speech emotion classification and simple text-to-speech generation tasks. For each task, we prepare the instruction, demonstrations (e.g., $\{x_1, y_1, x_2, y_2\}$), and the query $x_q$. The LLAMA 2 model is then asked to predict the corresponding result $y_q$. Here, $y_q$ can be either text or audio.

In this study, we propose a cross-modal in-context learning approach, empowering the frozen LLMs to solve any user-defined audio tasks based on a few demonstrations without any parameter update. To achieve this, we introduce a vector quantization audio codec model, named LLM-Codec, that maps the audio modality to the token space of frozen LLMs (*e.g.*, LLAMA 2 [36]). Our motivation is to reduce modality heterogeneity between audio and text by compressing audio data into the token space of LLMs. Given that the compressed audio and text modalities share a vocabulary, the compressed audio sequence can be treated as a new *foreign language*, which LLMs can learn from a few demonstration samples. Moreover, since LLMs are pre-trained on large-scale data and have discovered numerous token sequence patterns, they are well-positioned to generalize to this new *foreign language*. Figure 1 illustrates the integration of the proposed LLM-Codec with LLAMA 2 models for performing various audio tasks.

The proposed LLM-Codec aims to compress audio data into a lexical word sequence. A desired LLM-Codec should exhibit the following properties: (1) **Completeness** [12]: it should recover compressed audio with minimal loss. (2) **Compactness**: it should encode the audio into fewer-token sequences. (3) **Semantic richness**: it should encode audio into semantically rich token sequences, making them easier for pre-trained LLMs to recognize. Thus, we propose a semantic-guided multi-scale residual vector quantization (RVQ) based codec. Specifically, the codec model consists of three residual VQ layers: the first layer encodes semantic information, the second encodes coarse-grained acoustic information, and the third encodes residual acoustic information. Unlike previous works [9, 52], which encode audio data into the same granularity in each layer, we propose a multi-scale approach that encodes audio data at different granularities across layers. We are motivated by the observation that semantic-level information can be preserved with fewer tokens, while acoustic-level information requires more tokens This multi-scale approach not only shortens the token sequence length but also offers flexibility for various tasks; for instance, audio understanding tasks may only require the semantic-level VQ layer. Additionally, we design a novel semantic loss and consistency loss to enhance the training of the LLM-Codec model.

We conduct experiments to validate the effectiveness of LLM-Codec in an in-context learning setting. Using the pre-trained LLAMA 2 7B model without parameter updates, we evaluate LLM-Codec on various audio understanding and generation tasks, such as speech emotion classification, audio classification, simple text-to-speech, and speech denoising. The main contributions of this work are summarized as follows:

- We propose a novel LLMs-driven audio codec model, LLM-Codec, which effectively bridges the text and audio modalities. To the best of our knowledge, this is the first work to quantize the audio data into the representation space of LLMs.

- We demonstrate the feasibility and potential of using the in-context learning ability of LLMs to solve unseen audio tasks, including audio understanding and generation tasks. Extensive experiments and ablation studies further validate the effectiveness of our method.

## 2   Related works

**Audio Codec Models** Audio codec models have been widely used in the audio generation domain [50, 48, 5, 49, 46]. Historical investigations into low-bitrate parametric audio codecs began with earlier studies [21, 4]. However, the quality of these codecs typically faced limitations. Recently, advancements in neural network-based audio codecs have led to several promising developments [52, 9, 47, 25, 20]. These systems typically involve an encoder that extracts deep features from a latent space, which are then quantized and transmitted to a decoder for reconstruction. Particularly relevant to our work are the FACodec [20] and SpeechTokenizer [54] models, which explicitly model different properties of audio in different vector quantization layers. In contrast, our proposed LLM-Codec encodes audio data into a lexical word sequence, differing from these approaches.

**Multimodal Large Language Models** Recently, there has been tremendous progress in the area of multimodal LLMs. These models use pre-trained LLMs as the base and incorporate additional input modalities, such as vision [55, 28, 51, 56, 27, 37] and audio [7, 23, 15, 53, 35, 19, 40]. In general, multimodal LLMs consist of a pre-trained LLM, a pre-trained vision/audio encoder, and a modality adaptor. These systems often require the construction of extensive multimodal datasets to fine-tune the models. In the audio modality, most previous works focus on solving speech understanding [15] or general audio understanding [35, 19], but these models are not applicable to audio generation tasks. SpeechGPT addresses some audio understanding and generation tasks by fine-tuning all parameters and expanding the speech token vocabulary based on LLAMA. However, the speech tokens in SpeechGPT contain only semantic-level information, limiting its application to broader audio tasks, such as text-to-audio generation. Additionally, SpeechGPT does not explore the in-context learning ability to handle unseen tasks.

**In-context Learning** In-context learning, a form of few-shot learning, allows a large language model (LLM) to quickly adapt to a specific task during inference by reviewing just a few examples provided in the prompt [6]. This approach has proven successful in both natural language [45] and visual-language tasks [3, 51, 56, 28]. In the audio domain, several advanced methods have been proposed to leverage in-context learning for solving unseen audio tasks. SALM [7] introduces speech-augmented language models using in-context learning to address speech recognition and translation tasks, demonstrating that the SALM model can also handle keyword boosting. ICL-GSLM [13] employs warmup training and prompt tuning strategies to enhance the in-context learning abilities of pre-trained speech language models [26] for unseen tasks. However, ICL-GSLM primarily focuses on audio understanding tasks and overlooks audio generation tasks. Dynamic-superb [16] employs instruction-tuning for audio understanding tasks. Similarly, [42] and [41] investigate in-context learning within the speech understanding domain. Building on the success of in-context learning in NLP [45] and vision-language tasks [51, 56, 28], our study focuses on harnessing the in-context capabilities of frozen LLMs to address a broad range of audio understanding and generation tasks.

## 3   LLM-Codec

### 3.1   Overview

Previous audio codec models [9, 52, 47] use a VQ-VAE [38] framework, where the audio signal is first encoded into a discrete latent space and then decoded back into audio. Because audio codec models map the audio signal into discrete token sequences, many studies [17, 48, 5] have proposed training auto-regressive (AR) language models to generate these sequences, inspired by the success of LLMs in natural language processing. However, there is modal heterogeneity between the discrete audio tokens produced by the codec models and the text tokens used in LLMs. For example, the codebooks in audio codecs and the vocabularies of LLMs are typically unconnected, making it difficult to extend well-trained LLMs to audio modalities. Although previous works [53, 34] have demonstrated the effectiveness of expanding the vocabulary of LLMs to audio tokens and updating all of the parameters of LLMs, it will cost a lot of computing resources and forget the knowledge of the text. In this part, we present a large language models-driven audio codec model (LLM-Codec), which effectively bridges the gap between audio and text modalities. LLM-Codec is built on the VQ-VAE framework, but it differs from previous work in several key ways: (1) LLM-Codec is forced to quantize the audio signal into the token space of LLMs; (2) LLM-Codec adopts a multi-scale residual vector quantization strategy to balance the completeness and compactness of codec model;

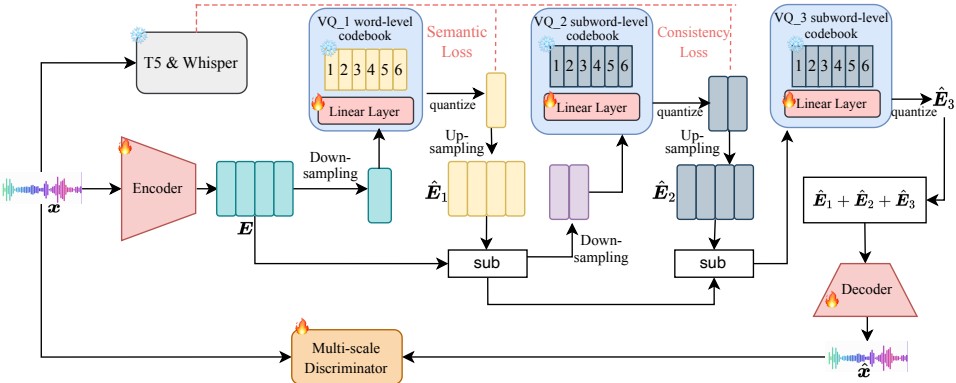

Figure 2: This figure provides a high-level overview of LLM-Codec, including an encoder, a decoder, a multi-scale discriminator, and multi-scale residual VQ layers. Here, 'sub' denotes feature subtraction. Note that the modules marked with a snowflake are frozen during training.

(3) LLM-Codec explicitly encodes different level information in different VQ layers. The following sections provide detailed insights into LLM-Codec, with Figure 2 offering a visual depiction.

## 3.2 Encoder and Decoder

For any audio $x$, the encoder first encodes it into latent presentations $E^{T,d}$, where $T$ denotes the number of frames, $d$ denotes the dimension of each vector. We set 4 down-sampling layers with $S = [3, 4, 5, 8]$ in the encoder, which results in 480 times down-sampling for audio. Then, each frame $e \in E$ is passed through the quantizer, which assigns it to the closest entry in a codebook, resulting in the quantized embedding $\hat{e}$. Finally, the quantized feature $\hat{E}$ inputs into the decoder to reconstruct $\hat{x}$. Note that we add several Transformer layers in both Encoder and Decoder part to maintain a good reconstruction performance. Refer to Appendix B to find more model structure details.

## 3.3 Multi-scale residual vector quantization with the vocabulary of frozen LLM

We use three residual VQ layers to maintain the balance between completeness and compactness. Furthermore, we propose to set different quantization granularity in different VQ layers: we expect the first VQ layer can encode the semantic information, and such information can be saved with fewer tokens, thus an interpolation function is used to down-sample the encoder features $E^{T,d}$ into $E_1^{T/k_1,d}$, then $E_1^{T/k_1,d}$ is passed through the first VQ layer to obtain $\hat{E}_1^{T/k_1,d}$. For the second VQ layer, we expect it can encode coarse-grained acoustic information, thus we pass the residual of the first VQ layer into the next VQ layer. Before that, we first up-sampling $\hat{E}_1^{T/k_1,d}$ into $\hat{E}_1^{T,d}$, then obtain the residual features by

$$E_2^{T,d} = E^{T,d} - \hat{E}_1^{T,d}. \tag{1}$$

Similarly, we also apply a down-sampling operation to $E_2^{T,d}$, we set the down-sampling step as $k_2$. The features become as $E_2^{T/k_2,d}$. Then we pass it into the second VQ layer and obtain $\hat{E}_2^{T/k_2,d}$. Lastly, we expect the last VQ layer can preserve all of the residual acoustic information. We first obtain the residual features based on the quantized features of the first two VQ layers

$$E_3^{T,d} = E^{T,d} - \hat{E}_1^{T,d} - \hat{E}_2^{T,d}. \tag{2}$$

Considering the residual acoustic information is more complex and diverse, we directly apply the VQ operation to each frame without any down-sampling. By using a large down-sampling step in the encoder of codec, and applying a multi-scale VQ strategy, we can effectively reduce the number of quantized audio token sequences. In our setting, 1-second audio with a 16k sampling rate will be quantized into 57 tokens. To ensure that the first VQ layers encode semantic information, we propose incorporating a semantic loss during the training process. Furthermore, to maintain the

Table 1: Performance comparison between open-sourced audio codec models, baselines, and the proposed LLM-Codec. * means the reproduced results by ourselves.

| Model | Down-sampling | Tokens per second | PESQ | STOI | SFTF loss |
|---|---|---|---|---|---|
| Encodec_24k (3 Vanilla RVQ) [9] | 320 | 225 | 2.18 | 0.79 | 1.21 |
| DAC_16k (3 Vanilla RVQ) [25] | 320 | 150 | 1.76 | 0.78 | 1.43 |
| Baseline* (3 Vanilla RVQ) | 480 | 99 | 2.64 | 0.83 | 1.09 |
| Baseline* (2 Multi-scale RVQ) | 480 | 41 | 2.22 | 0.79 | 1.23 |
| Baseline* (1 Vanilla VQ) | 480 | 33 | 2.01 | 0.76 | 1.26 |
| LLM-Codec (Ours) | 480 | 57 | 2.55 | 0.82 | 1.15 |

training stability, we propose a consistency loss. The details will be introduced in Section 3.4.

**The initialization of VQ layers** To generate lexical tokens, we utilize a pre-trained LLAMA 2 codebook to initialize the VQ layers. Considering that the first layer, the VQ layer, is designed to encode the semantic information, we do not directly use the full LLAMA codebook. Instead, we define a new codebook based on Oxford 5000 Words, these words are commonly used to make up any meaningful sentence. We choose these words that only consist of one or two sub-words in the LLAMA codebook. If a word includes two sub-words, we use the mean representation of two sub-words in the LLAMA codebook as the final representation. Lastly, the codebook size of the first VQ layer is 3248. We directly use the LLAMA codebook to initialize the second and third VQ layers. The codebook size is 32000. Furthermore, the LLAMA codebook embedding dimension is 4096, which is too large for codec training. Thus, we apply a linear mapping to 512. In the training process, the parameters of codebooks are fixed.

## 3.4 Training loss

Our approach is based on a GAN objective, in which we optimize both the generator(it consists of encoder, quantizer, and decoder) and the discriminators. For the generator, its training loss consists of three parts: (1) reconstruction loss term; (2) adversarial loss term (via discriminators); and (3) semantic and consistency losses. In the following, we give the details of proposed semantic loss and consistency loss. Refer to Appendix B.2 to find the details of reconstruction loss and adversarial loss.

**Semantic loss** To enhance the semantic representation ability in the first layer, we introduce a semantic loss for the first VQ layer. We expect it can encode semantic information, for example, if the input audio includes a sound event, the first layer should encode which semantic information of the sound event. Similarly, if the input audio is speech, the first layer should encode the content of the speech. To realize this target, we use a pre-trained T5-base model [32] to extract a global representation vector $g$ for the input audio content. We use Whisper to obtain its transcriptions if the input audio is speech. If the input audio is sound, we use its audio caption label:

$$\mathcal{L}_s = L_1(mean(\hat{\boldsymbol{E}}_1^{T,d}), \boldsymbol{g}) \tag{3}$$

**Consistency loss** In our early experiments, we found the training of LLM-Codec is not stable, and the model is easy to collapse. One of the reasons is that we designed a significant down-sampling rate and the codebooks are fixed in the training, which increases the training difficulty. To solve this issue, we propose a consistency loss to maintain the training stability. Specifically, we propose using a pre-trained Whisper encoder [31] to extract frame-level features $\boldsymbol{w}$, then using these features as prior knowledge to guide the second VQ layer.

$$\mathcal{L}_c = L_1(\hat{\boldsymbol{E}}_2^{T/2,d}, inp(\boldsymbol{w})) \tag{4}$$

where $inp$ denotes the interpolation function to align the feature dimension between the quantized features and whisper features. We chose the Whisper encoder because it is trained not only on speech data but also on non-speech data. Furthermore, we do not apply this loss on the third VQ layer, because we expect the third VQ layer to encode the residual information.

## 4 UniAudio 1.5

By combining the pre-trained LLMs and the proposed LLM-Codec models, we can solve many audio tasks in a few-shot style, as Figure 1 shows. We named the system UniAudio 1.5 for the reason that

Table 2: Audio understanding task evaluation results. Task induction denotes the explanatory text that precedes the sequence of audio and text. It is intended to describe the task to the model in natural language, for example: Please answer the question. Accuracy (%) is used as the metric. For the Random guess, we calculate the average based 5 times evaluation. K shots refers to the number of distinct samples for each category, and Repeats refer to how many times we copy the prompt samples.

| Method | # Layers | Task Induction | ✗ | ✓ | ✓ | ✓ | ✓ |
| | | K Shots | 1 | 1 | 3 | 1 | 1 |
| | | Repeats | 0 | 0 | 0 | 2 | 3 |
|---|---|---|---|---|---|---|---|
| *2-way speech emotion classification* | | | | | | | |
| Random | None | | | | 44 | | |
| BLSP [40] | Whisper encoder | | 9 | 29 | 50 | 33 | 19 |
| LLM-Codec | semantic layer | | 25 | **53** | **59** | 53 | 54 |
| LLM-Codec | semantic + acoustic layers | | **45** | 49 | 53 | **55** | **54** |
| *2-way sound event classification.* | | | | | | | |
| Random | None | | | | 45 | | |
| BLSP [40] | Whisper encoder | | 44 | 47 | 54 | 15 | 17 |
| LLM-Codec | semantic layer | | **48** | **60** | 57 | 57 | **73** |
| LLM-Codec | semantic+acoustic layers | | 41 | 48 | 55 | 54 | 62 |
| *3-way sound event classification.* | | | | | | | |
| Random | None | | | | 30 | | |
| BLSP [40] | Whisper encoder | | 23 | 26 | 36 | 24 | 16 |
| LLM-Codec | semantic layer | | **38** | **41** | **39** | 43 | 42 |
| LLM-Codec | semantic+acoustic layers | | 25 | 37 | 35 | **44** | **50** |

the system can be viewed as a universal audio task solver.

**Connection to UniAudio** UniAudio 1.5 is an advanced edition of the UniAudio Series [48]. Compared to its previous version UniAudio [48], UniAudio 1.5 has the following connections and distinctions. First, **goal**. While both UniAudio 1 and UniAudio 1.5 aim at building a universal audio foundation model for all audio tasks, their focuses are different. UniAudio focuses on audio generation tasks, such as text-to-speech, text-to-music, singing voice generation, and so on. UniAudio 1.5 focuses on audio understanding and generation tasks by exploring the few-shot ability based on large language models. Second, **architecture**. UniAudio 1.5 keeps the basic components in UniAudio, such as an audio codec used to transfer the audio modality into discrete representations, and a decoder-only transformer backbone is used. However, UniAudio 1.5 leverages 1) a pre-trained LLMs to solve the audio understanding and generation tasks by in-context learning, 2) an LLM-driven audio codec to quantize the audio data into the token space of LLMs. Building a multi-modal audio foundation model that is capable of handling any audio task is the ultimate goal of the UniAudio series. In UniAudio 1.0, we show the possibility of building a universal model for different types of audio generation tasks, but it (1) cannot effectively solve audio understanding tasks and (2) cannot solve unseen audio tasks in the training or fine-tuning stages. UniAudio 1.5 shows the possibility of using pre-trained LLMs for both audio understanding and generation tasks. We believe the proposed LLM-Codec in UniAudio 1.5 builds a foundation for more advanced editions of the UniAudio Series in the future.

## 5 Experimental Results

### 5.1 Experimental Settings

#### 5.1.1 LLM-Codec training and reconstruction performance evaluation

**Training data** LLM-Codec is a universal audio codec model, trained on both speech and sound datasets. For speech data, we use a portion of the MLS dataset [30], and for sound data, we use the AudioCaps dataset [22]. In total, we utilized 2k hours of audio data to train the LLM-Codec model.
**Model setting** The details of the LLM-Codec model configuration can be found in Appendix B. For the proposed multi-scale RVQ, we set three scales, with down-sampling rates of $k_1 = 4$, $k_2 = 2$, and $k_3 = 1$ for each layer. We initialize the VQ layers using the vocabulary of the LLAMA 2 7B model.

Table 3: Evaluation on dynamic-superb benchmark tasks. Accuracy (%) is used as the metric.

| Task | ImageBind-LLM [16] | Whisper-LLM [16] | ASR-ChatGPT [16] | Ours |
|------|--------------------|--------------------|--------------------|------|
| Accent Classification | 19 | 4 | 7 | **24** |
| Bird Sound Detection | 28 | 14 | 15 | **50** |
| Chord Classification | 44 | **58** | 3 | 55 |
| Language Identification | 26 | 13 | **96** | 25 |

The VQ layers are fixed during the training stage.

**Evaluation metrics** To verify the reconstruction performance, we use Perceptual Evaluation of Speech Quality (PESQ), Short-Time Objective Intelligibility (STOI), and Mel reconstruction loss.

**Evaluation data** We conduct evaluation on speech and sound datasets, for speech data, we choose 200 utterances from VCTK [39], and for sound data, we choose 200 utterances from ESC 50 dataset.

**Baselines** We compare our model with publicly available audio codec models, including Encodec [9] and DAC [25]. Additionally, we compare it with our reproduced audio codec models, which were trained on the same dataset as LLM-Codec.

### 5.1.2 LLMs equipped with the proposed LLM-Codec for downstream audio tasks

**Evaluation dataset** We choose commonly used test datasets for each task and construct N-way-K-shot test pairs. More details about constructing evaluation samples can be found in Appendix C.1.

**Baselines** Given the limited number of works focusing on few-shot learning for unseen audio tasks, we choose BLSP [40] as one of the baselines for audio understanding tasks. Since BLSP is fine-tuned on a continuation writing task and does not explicitly address audio classification, these tasks are considered unseen for the BLSP model. Additionally, we compare our approach with instruction-tuning-based models as described in dynamic-superb [16]. For audio generation tasks, we did not find directly related works, so we report the performance of state-of-the-art specialized models.

### 5.2 Main results

We first present the reconstruction performance comparison. Then we apply the LLM-Codec and LLAMA 2 7B model for audio understanding and audio generation tasks, to verify the ability of the proposed method. Lastly, we give the visualization of LLM-Codec to explain why it can work. We leave more experiments on Appendix D.

**Reconstruction performance** We compare the audio reconstruction quality with previous works Encodec [9], DAC-Codec [25], and our baseline model. We report Perceptual Evaluation of Speech Quality (PESQ) and Short-Time Objective Intelligibility (STOI). Table 1 shows the results. Compared to previous methods, the LLM-Codec achieves better reconstruction performance while utilizing fewer tokens. More specifically, the LLM-Codec model can compress 1-second audio data into a sequence that only includes 57 tokens, which significantly reduces the sequence length. Compared to the RVQ baseline model, the LLM-Codec significantly reduces the compressed tokens, and its reconstruction performance does not significantly decline. In Section 5.3, we will show the importance of compressing audio into fewer tokens. We also conduct experiments to validate whether we can use a few VQ layers, such as 1 VQ layer or 2 VQ layer, we can see that the reconstruction performance will significantly drop. To maintain the balance between completeness and compactness, we choose a multi-scale 3 VQ layer as the default setting.

**Speech Emotion Classification** The speech emotion classification task [10] aims to predict the emotion label of the speech. We conduct 2-way K-shot experiments on the ESD [1] dataset. Experimental results are shown in Table 2. We have the following findings: (1) Task induction is important to maintain the stability of performance, we can see that without task induction, the classification accuracy will dramatically decline. (2) The semantic layer effectively extracts

Table 4: Text-to-speech generation performance.

| Model | ACC | DNSMOS |
|-------|-----|--------|
| GT | - | 2.91 |
| FastSpeech 2 | - | 3.42 |
| LLM-Codec (Ours) | 70 | 2.92 |

the global semantics of audio, which can be easily understood by the LLAMA model. (3) Using more demonstration samples (*e.g.* 3 shots), the performance will be better. (4) Repeating the demonstration

samples can also bring improvement. (5) Compared to the BLSP, our method performs better in any setting. Furthermore, we also note that the performance of BLSP will drop when repeat operation is used. One possible reason is that BLSP only learns the translation relationship between text and speech, repeating samples cannot bring new cues for LLMs to solve the new task. Instead, our LLM-Codec learns to map the audio data into the latent space of LLMs, increasing the number of demonstration samples can help LLMs to find special patterns to solve this new task.

**Sound Event Classification** Sound event classification aims to recognize the sound event in the audio. In general, an audio may include multiple events. To simplify the recognition difficulty, we assume each audio only includes one event. We conduct experiments on the ESC50 dataset [29], which includes 50 different types of events. We construct 2-way-K-shot and 3-way-K-shot evaluations based on the ESC50 test set. Compared with the BLSP model, our proposed method gets better performance. Based on the experimental results from two audio understanding tasks, we can see that the semantic VQ layer is very important for understanding tasks.

**Dynamic-SUPERB Benchmark** We also conduct experiments on Dynamic-SUPERB Benchmark tasks [16]. In [16], authors propose an instruction-tuning strategy for multi-modal LLMs. They first construct a lot of audio tasks as training data, then validate some unseen audio tasks in a zero-shot way. To make a fair comparison, we use the same test set with them, and choose the first N samples as the demonstration to construct a N-way-1-shot evaluation. As Table 3 shows 4 selected audio understanding tasks, our proposed method obtains better or compared performance than these baselines in [16]. Especially, for the bird sound detection task, our proposed method obtained great improvement over previous methods. We also note that our method performs worse on language identification, the possible reason is that our codec model is only trained on English speech data. In the following, we will show that LLM-Codec also can be used to conduct audio generation tasks.

**Simple text-to-speech generation** We conduct text-to-speech generation on the Free Spoken Digit Dataset (FSDD) dataset [11], which includes 3 speakers and 1,500 recordings. Unlike the traditional TTS model, which generates any speech content, this task generates digit speech. Our current model to generate complex speech content is still challenging. We use accuracy (ACC) to assess the content of the generated sample whether following the instructions. DNSMOS is used to assess the speech quality of generated samples. We construct 20 different query questions, including addition, subtraction, multiplication, division, and reasoning (finding more details from Appendix C.2). From Table 4, we can see that our proposed model can accurately understand the query in most cases (the accuracy is 70 %) and generate good-quality speech samples. Figure 3 gives a visualization of generating speech based on the query. The frozen LLAMA model learns about 4 digits (0-3), each audio digit includes 5 samples. We add the context for each audio: "an audio of k" before inputting the audio's discrete representations into LLAMA, as Figure 1 (b) shows. After that, we let the LLAMA 2 model generate corresponding speech digits based on the instruction. We also note that the generated audio appears different from all context audio samples, demonstrating the cross-modal reasoning capability of LLMs when using the LLM-Codec as the connector for text and audio.

**Simple speech denoising** To verify whether the proposed method can conduct speech-denoising tasks, we simulate noisy speech based on the VCTK dataset and NoiseX-92 dataset, we set the SNR ranges from -20 to 20. For each clean speech, we choose 5 different noises to simulate noisy speech. The first 4 noisy and clean audio pairs are used as demonstrations, and the model learns to denoise the last noisy one. To improve in-context learning ability, we repeat the demonstration

Table 5: Speech denosing evaluation.

| Model | PESQ | STOI |
|---|---|---|
| SGMSE+ [33] | 3.53 | 0.79 |
| LLM-Codec (Ours) | 2.17 | 0.57 |

samples 4 times. The experimental results as Table 5 shows, we can see that the proposed method can also learn to denoise without any training. Furthermore, we also note that the performance has a large room to improve compared to special models.

**Token Visualization** We visualize the tokens produced by the first VQ layer of LLM-Codec for different types of sound in Figure 4. We have the following findings: (1) Although the two audios include the same sound event, the quantized sequence is not exactly the same. (2) The quantized sequence of two same types of audio has a similar pattern, *e.g.* their token sequences have similar repeating patterns or the same word. Such patterns may help the LLMs recognize the type of audio.

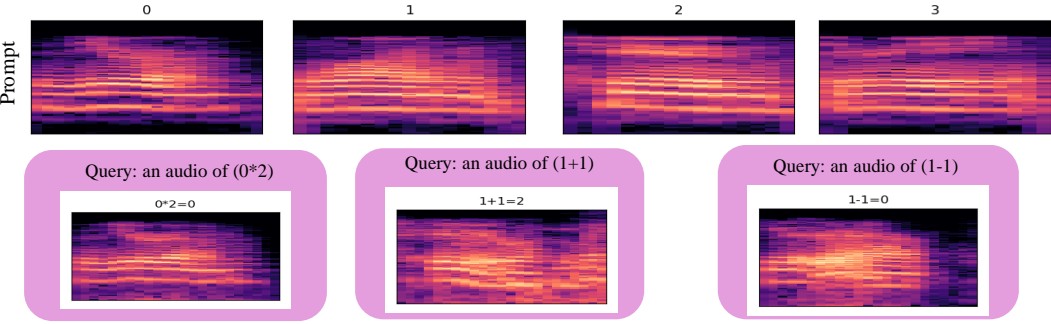

Figure 3: Examples of simple text-to-speech generation using LLM-Codec and LLAMA2 model.

## 5.3 Ablation study

**The influence of multi-scale RVQ** We first conduct experiments to see the effectiveness of multi-scale RVQ. As Table 6 shows, compared with vanilla RVQ, the proposed multi-scale RVQ does not bring a significant reconstruction performance drop, which validates our assumption that semantic information does not need too much token to encode. Secondly, we find the multi-scale RVQ significantly reduces the length of the token sequence and brings benefits for downstream tasks (the audio classification accuracy is better than the baseline). One potential explanation is that LLMs can better identify the unique pattern in a brief sequence. Intuitively, the semantic information included in a 1-second audio is limited. It is unnecessary to use very long sequences to represent limited information.

**The influence of down-sampling times** We can see that using a smaller down-sampling rate (320) can improve the reconstruction performance, but it also increases the length of the token sequence. We can see that the classification accuracy will decrease when the sequence length increases.

**The influence of semantic loss** Without semantic loss, the performance of the audio understanding task will drop. Furthermore, we also find that adding semantic loss does not influence the reconstruction performance. In summary, the proposed semantic loss is very useful.

**The influence of consistency loss** We find that consistency loss is important to maintain training stability. Without it, we can see the model fails to reconstruct the audio. We conjecture that frozen codebooks and large compression rates significantly improve the difficulty of training. The consistency loss forces the second VQ layer to produce features similar to those of the Whisper encoder, which provides guidance for vector quantization and prevents the model from collapsing in the early stage.

**The influence of word-level codebooks** We also conduct experiments to show the effectiveness of using word-level codebooks to initialize the first VQ layer. Compared with using sub-word vocabulary for the first VQ layer, we can see that using the proposed word-level codebook can improve the reconstruction performance and classification accuracy.

**The importance of frozen codebooks** LLM-Codec compresses the audio data into the token space of LLMs by initializing the codebooks with the LLMs' vocabulary and fixing it during the training stage. Table 6 also presents the results of updating codebooks: it can improve the reconstruction performance, but the accuracy is a significant drop. The result is consistent with our hypothesis: updating the codebooks parameter will decrease the codec training difficulty, but the learned codebook space is different from the LLM's token space, resulting in the downstream task performance declines.

**Different setting of $k_1$ and $k_2$ in multi-scale RVQ** We validate a new setting for multi-scale RVQ with $k_1 = 3$ and $k_2 = 5$. We can see that the reconstruction performance will decline. We think one of the reasons is that the second VQ layer should not apply a large down-sampling step, which significantly influences the reconstruction.

**Codebook usage** Previous works [18, 52, 25] suggest that using a large-scale codebook may result in codebook collapse ((where a fraction of the codes are unused). We calculate the codebook usage for each VQ layer in LLM-Codec. The used codes are 3246 (3248), 31911 (32000), 31941 (32000) for each VQ layer, which shows that most of codes are used.

Table 6: Ablation studies on training loss, multi-scale RVQ setting, initialization of VQ layer. The classification accuracy (%) is evaluated under the sound event classification task 2-way 1-shot setup.

| Model | Down-sampling | Tokens per second | PESQ | STOI | ACC |
|---|---|---|---|---|---|
| Baseline (3 Vanilla RVQ) | 480 | 99 | 2.64 | 0.83 | 55 |
| LLM-Codec (3 Multi-scale RVQ) | 480 | 57 | 2.55 | 0.82 | 60 |
| LLM-Codec (3 Multi-scale RVQ) | 320 | 87 | 2.60 | 0.83 | 57 |
| w/o semantic loss | 480 | 57 | 2.54 | 0.82 | 58 |
| w/o consistency loss | 480 | 57 | 1.19 | 0.53 | 48 |
| w/o word-level codebook | 480 | 57 | 2.46 | 0.81 | 59 |
| updating codebooks | 480 | 57 | 2.63 | 0.83 | 55 |
| seting $k_1 = 3$ and $k_2 = 5$ | 480 | 50 | 2.35 | 0.79 | 58 |

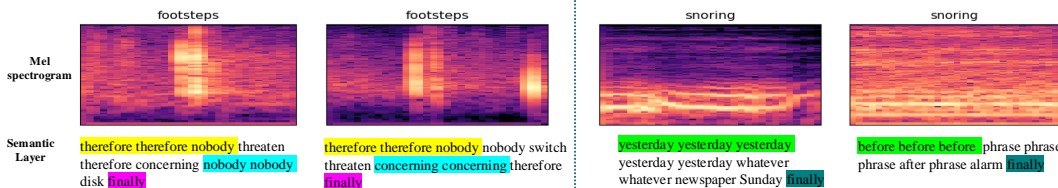

Figure 4: The token visualization of the semantic layer of LLM-Codec is shown. We present two groups of samples, each containing two audio recordings with the same sound event label. In each group, we use the same color to highlight potentially similar patterns in the two audio recordings, such as identical token sub-sequences or token repeating frequencies. We speculate that these patterns can be easily recognized by LLMs, allowing them to learn new sound events quickly with just a few demonstrations.

# 6   Conclusion

In this study, we explore a cross-modal in-context learning approach to solve unseen audio tasks in a few-shot style. Specifically, we propose to train a LLMs-driven audio codec (LLM-Codec) that compresses the audio signal into the token space of LLMs. The LLM-Codec effectively reduces the modal heterogeneity between text and audio. With the help of LLM-Codec, pre-trained LLMs can be applied to solve multiple audio understanding and generation tasks. We demonstrate that LLM-Codec has good reconstruction performance, and the compressed token sequence is suitable for LLMs to understand and generate. Experiments show that the LLMs equipped with the proposed LLM-Codec, named as UniAudio 1.5, prompted by only a few examples, are capable of achieving the expected functions in many scenarios.

# 7   Limitations

Although we show the possibility of using the in-context learning ability of LLMs for unseen audio tasks without any parameter update, the performance of these tasks is still poorer than these special models in the audio domain. The capability to learn within an in-context framework is significantly limited for a modality that was not exposed during the training process. Due to the LLM's context length limitation, we cannot add more demonstration samples to help improve the performance. We think it is worth exploring using more demonstrations to improve its in-context learning ability. Moreover, we only explore the use of the LLAMA 2 7B model as the backbone; more advanced open-sourced LLMs are worth exploring. Considering we have open-sourced the LLM-Codec, readers can conduct experiments on their favorite LLMs. Furthermore, there are fewer theoretical connections to justify the meaning of the lexical representation of the "trainable new (pseudo) language" of speech, which can be improved. In future work, we will explore to train multi-modal LLMs by fine-tuning LLMs on text-audio datasets with the help of LLM-Codec, and build more theoretical analysis.

## 8 Acknowledgements

This work is partially supported by the Centre for Perceptual and Interactive Intelligence (CPII) Ltd., a CUHK-led under the InnoHK scheme of Innovation.

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

# Appendices

## A  Appendix Overview

These Appendices provide additional details to support our main manuscript, including (1) the training detail and model structure of LLM-Codec. (2) The details of the evaluation dataset. (3) More audio task evaluation results. (4) Limitations.

## B  More details of LLM-Codec

### B.1  Model structure

Table 7 gives the details of LLM-Codec configuration, which results in 160M parameters. To facilitate research on cross-modal in-context learning and multi-modal LLMs, we have open-sourced the LLM-Codec models.

|  | LLM-Codec |
| --- | --- |
| Input shape | (1, 1, T) |
| Encoder (input dimension) | 32 |
| Down-sampling rate | [3, 4, 5, 8] |
| latent dimension | 512 |
| Codebook dimension | 4096 |
| Transformer layer dimension | 512 |
| Number of Transformer heads | 8 |
| Decoder dimension | 1536 |
| Up-sampling rate | [8, 5, 4, 3] |
| VQ strides | [5, 3, 1] |

Table 7: LLM-Codec model backbone configurations

**Encoder and Decoder** Considering a single-channel audio signal $x \in \mathcal{R}^{t \times sr}$, where $t$ and $sr$ denote the audio duration and the sample rate. The overall architecture is similar to previous audio codec models, such as Encodec [9], DAC [25], and HiFi-Codec [47], which includes four main parts: encoder, quantizer, decoder, and discriminators. Figure 2 provides a visual depiction of the proposed method. For any input $x$, the encoder first encodes it into latent presentations $\boldsymbol{E}^{T,d}$, where $T$ denotes the number of frames, $d$ denotes the dimension of each vector. Due to the encoder includes some down-sampling layers, resulting in $T << t \times sr$. Then each frame $e \in \boldsymbol{E}$ is passed through the quantizer, which assigns it to the closest entry in a codebook, resulting in the quantized embedding $\hat{e}$. Finally, the quantized feature $\hat{\boldsymbol{E}}$ inputs into the decoder to reconstruct $\hat{x}$. The encoder and decoder architecture follows previous works Encodec [9] and DAC-Codec [25], which includes several convolution layers and transformer layers. Specifically, the encoder model comprises a 1D convolution with $C$ channels and a kernel size of 7, leading into $B$ convolution blocks. Each block contains a residual unit followed by a down-sampling layer, which employs a convolution with a kernel size $K$ that is twice the stride $S$. The residual unit itself comprises two convolutions, each with a kernel size of 3, linked by a skip connection. The transformer block is used for sequence modeling, and concludes with a final 1D convolution layer featuring a kernel size of 7. In this study, we set $S = [3, 4, 5, 8]$, which results in 480 times down-sampling for audio. The decoder mirrors the encoder's architecture, substituting stride convolutions with transposed convolutions and reversing the stride order.

**Discriminators** For the discriminators, we follow previous work [48], which combines the mel-spectrogram and log-mel-spectrogram features and then input them into a network consisting of several convolutional layers. In our experiments, we use 6 different discriminators with different configurations. Specifically, we set the hidden dimension as {64, 128, 256, 512, 512, 512} and the hop length as {32, 64, 128, 256, 512, 1024}.

## B.2 Reconstruction loss and adversarial loss for LLM-Codec

The reconstruction loss is calculated between $x$ and $\hat{x}$. We design the loss from two aspects: the time domain and the frequency domain. For the time domain, we directly calculate the $L_1$ loss between $x$ and $\hat{x}$. For the frequency domain, we calculate the $L_1$ loss between the STFT spectrogram of $x$ and $\hat{x}$. Note that a sub-band split strategy [43] is used to split the spectrogram into several parts, and then we calculate the loss between these sub-bands. The adversarial loss is used to improve the perceptual quality of generated audio. A multi-scale Mel-spectrogram discriminators [48] is used. To train the discriminator, we can optimize the following objective function:

$$\mathcal{L}_d = \frac{1}{K} \sum_{i=1}^{K} max(0, 1 - D_k(\boldsymbol{x})) + max(0, 1 + D_k(\hat{\boldsymbol{x}})) \tag{5}$$

where $K$ denotes the number of discriminators. In the training stage, the adversarial loss for the generator is calculated as a hinge loss over the logits of these discriminators:

$$\mathcal{L}_{adv} = \frac{1}{K} \sum_{i=1}^{K} max(0, 1 - D_k(\hat{\boldsymbol{x}})) \tag{6}$$

We also compute the feature loss by taking the average absolute difference between the discriminator's internal layer outputs for the generated audio and those for the corresponding real audio.

## B.3 Training details

The AdamW optimizer is used in the training. We set the learn rate as $1e - 4$. We train the model with 100k steps. For the training loss, we combine all of the loss terms without a special loss design. In the training stage, we use the pre-trained T5-base model and Whisper-base model for the reason that their latent dimension is both 512. We conduct all of the experiments with 2 NVIDIA A100-80G GPUs.

# C   Evaluation dataset

In this part, we show how to construct an evaluation dataset for the N-way-k-shot test.

## C.1   N-way-k-shot test samples

**Speech emotion classification with LLAMA 2.**   We give an example of 2-way 1-shot classification tasks. Firstly, we get the emotion class set from the ESD dataset: ['Angry', 'Happy', 'Neutral', 'Sad', 'Surprise']. Then we randomly choose two emotions as targets, and get the corresponding audios. For example, assuming that we get *Happy* and *Sad* the prompt can be

```
For each of the following input-output pairs, the output is
one of ['Happy' or 'Sad']
###
Input: <token sequence from a happy emotion of audio>
Output: happy
###
Input: <token sequence from a sad emotion of audio>
Output: sad
###
Input: <token sequence from the query audio>
Output:
```

We use greedy decoding to get a maximum of 16 tokens from LLAMA 2 7B.

**Sound event classification with LLAMA 2.**   We give an example of 3-way 1-shot classification tasks. Firstly, we get the sound event class set from the ESC50 dataset. Then we randomly choose three sound events as targets, and get the corresponding audio. For example, assuming that we get *dog*, *speaking*, and *mouse click* we set the prompt as

```
For each of the following input output pairs,
output is one of ['dog' or 'speaking' or 'mouse_click']
###
Input: <token sequence from a dog event of audio>
Output: dog
###
Input: <token sequence from a speaking event of audio>
Output: speaking
###
Input: <token sequence from a mouse click event of audio>
Output: mouse click
###
Input: <token sequence from the query audio>
Output:
```

## C.2   Audio generation

### Text-to-speech generation with LLAMA 2

```
Instruction: Learn a foreign language for different digits,
then generate the corresponding number using
foreign language based on instruction
###
Input: <an audio of 1>
Output: <token sequence of audio 1>
###
Input: <an audio of 2>
Output: <token sequence of audio 2>
###
Input: <an audio of 3>
Output: <token sequence of audio 3>
###
Input: <an audio of 1+1>
Output:
```

To simplify to generation process, we set each audio has the same duration.

**Text-to-speech question design**   we designed 20 different questions for text-to-speech, which include addition, subtraction, multiplication, division, and reasoning.

```
###
Input: <an audio of (1+1)>
###
Input: <an audio of (1+2)>
###
Input: <an audio of (2+2)>
###
Input: <an audio of (5-1)>
###
Input: <an audio of (5-2)>
###
Input: <an audio of (1-1)>
###
Input: <an audio of (0*2)>
###
Input: <an audio of (2*2)>
###
Input: <an audio of (1/1)>
###
Input: <an audio of (2/1)>
```

```
###
Input: <an audio of (4/2)>
###
Input: <an audio of (the square root of 4)>
###
Input: <an audio of (the square root of 1)>
###
Input: <an audio of (the last digit of 110)>
###
Input: <an audio of (the first digit of 110)>
###
Input: <an audio of (the sum of 1+1+1)>
###
Input: <an audio of (the next digit of 4)>
###
Input: <an audio of (sequence 0,1,2,3 what is next?)>
###
Input: <an audio of (sequence 4,3,2,1 what is next?)>
###
Input: <an audio of (how many days in a week)>
```

# D   More audio tasks evaluation experiments with the proposed method

In the following, we show the results of speech command recognition and text-to-sound generation.

## D.1   Speech Command Recognition

Table 8: Speech command recognition evaluation results on Speech Command dataset. Accuracy (%) is used as the metric. For the Random guess, we run 5 times then calculate the average.

| Method | # Layers | Task Induction | ✗ | ✓ | ✓ | ✓ | ✓ |
|--------|----------|----------------|---|---|---|---|---|
|        |          | K Shots | 1 | 1 | 3 | 1 | 1 |
|        |          | Repeats | 0 | 0 | 0 | 1 | 3 |
| LLM-Codec | semantic layer | | 50 | 53 | 59 | 54 | 56 |
| LLM-Codec | semantic+acoustic layers | | 25 | 53 | 58 | 49 | 52 |
| BLSP [40] | Whisper encoder | | 29 | 65 | 84 | 69 | 59 |
| Random | None | | | | 44 | | |

Speech command recognition refers to the recognition and interpretation of short phrases or keywords that are typically used to control devices or applications. In this part, we choose audio samples from the Speech Command dataset [44]. We choose four types of commands, including down, go, left, and right. For each command, we randomly choose 20 utterances, then we use these data to construct a 2-way-K-shot evaluation. Experimental results are shown in Table 8, we can see that only using the semantic VQ layer brings the best performance. Instead, if the tokens from the acoustic layer are used, the performance will decline. One possible reason is that for the speech command recognition task, it only needs to understand the content, and the content information has been saved in the semantic layer, the additional acoustic information may disturb the LLMs's prediction.

## D.2   Simple text-to-sound generation

Similarly, we can also use the same setting as text-to-speech to conduct text-to-sound generation tasks. We choose a test set from the ESC50 dataset [29], and let the model learn to generate sound events based on the text label. For example, we can set several different sound types in the prompt, and then ask the LLAMA model to generate a new audio. However, we also find that it is hard to ask LLM to generate new types of sound.

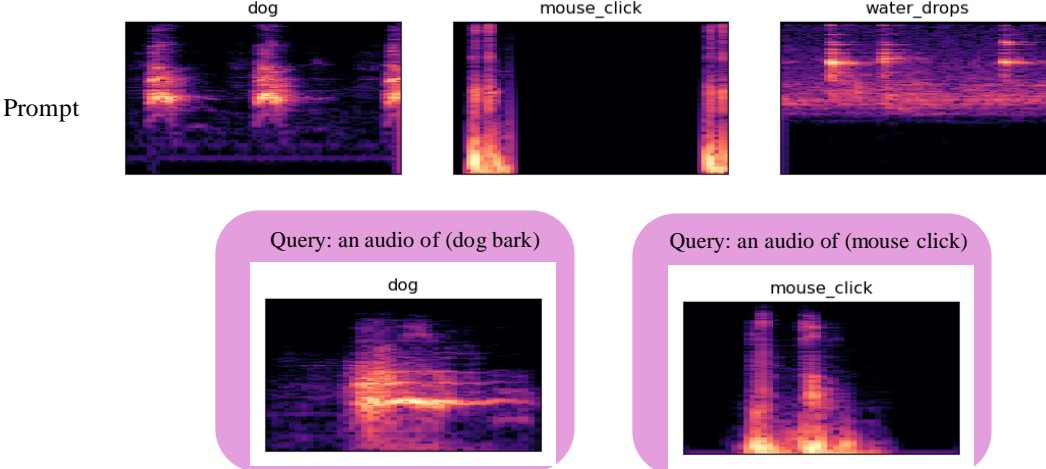

Figure 5: Examples of simple text-to-sound generation on FSDD dataset using LLM-Codec with a frozen LLAMA2 7B model.

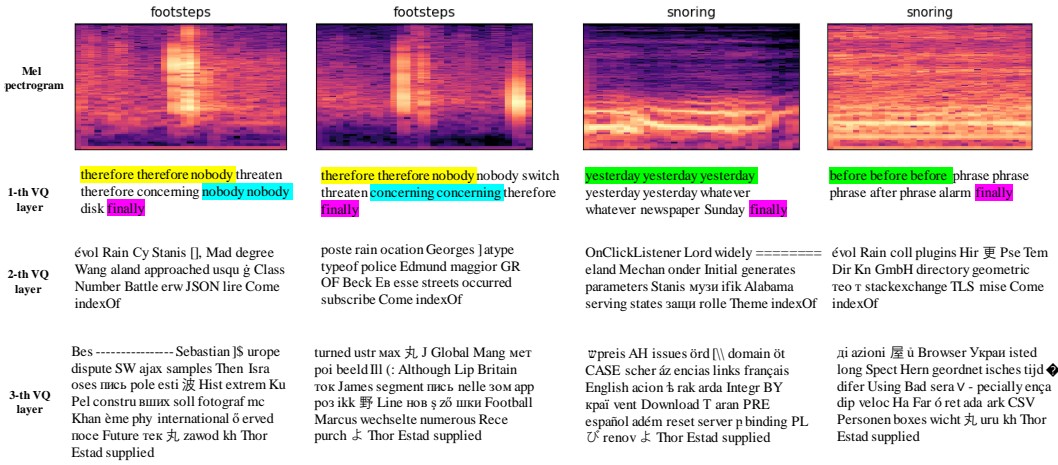

Figure 6: The token visualization of three VQ layers with LLM-Codec. The audio samples are from the ESC50 dataset.

### D.3 Token visualization

Figure 6 shows the details of three VQ layers token visualization. We have the following findings: (1) The few tokens in the first layers seem to more easily understand the audio's pattern. For example, we can easily find two audios with the same sound event that can be quantized into very similar sequences. Because we force the first VQ layer to encode the semantic level information. Instead, the second and third VQ layers aim to encode the acoustic information, but these audios have obvious differences in acoustic condition (we can observe it from its mel-spectrogram). Second, it is worth noting that all of the training data is English-related, but we can see that the encoded sequence also includes other languages, such as Chinese.

# E   Additional experiments in the rebuttal period

## E.1   Is there any scaling effects of the backbone LM selection?

Inspired by the reviewer's suggestion, we added an experiment to explore the influence of scaling effects of the backbone LM. Specifically, we compare the performance of different LM selections: LLAMA2 7B and LLAMA 2 13B. We conduct experiments on N-way-1-shot sound event classification, The performance comparison as Table 9 shows. We can see that scaling the backbone LM can also bring improvement for audio tasks.

Table 9: The influence of scaling effects of the backbone LM.

| Model / task | 2-way-1-shot | 3-way-1-shot | 4-way-1shot | 5-way-1-shot | 6-way-1-shot |
|---|---|---|---|---|---|
| LLAMA 7B | 60 | 41 | 36 | 33 | 17 |
| LLAMA 13B | 62 | 0.42 | 41 | 43 | 31 |
| BLSP | 47 | 26 | 15 | 12 | 10 |

## E.2   The performance comapred to other text-to-audio models.

We add a text-to-audio evaluation. Specifically, we choose previous SOTA AudioGen [24] as one of the baselines, because AudioGen is also an autoregression model based on audio codec models. Furthermore, we also choose some diffusion-based audio generation models, including AudioLDM and Tango, as the other baselines. For AudioLDM and Tango, we use their official checkpoints, and we set 200 diffusion steps for the inference. We conduct experiments on the ESC 50 [29] validation set. AudioGen, AudioLDM, Tango, and our model do not see the ESC 50 dataset in the training stage. We use the event label to construct the text description, e.g. if the event label is 'clapping', we will construct the caption as 'this is the sound of clapping.'. For the evaluation metrics, we follow previous works to use FAD and KL as the metrics. The results are shown in the table 10.

Table 10: The text-to-audio evaluation.

| model | FAD | KL |
|---|---|---|
| AudioGen | 20.4 | **1.94** |
| AudioLDM | 15.6 | 3.52 |
| Tango | **12.7** | 3.01 |
| Ours | 17.16 | 3.05 |

## E.3   The reconstruction performance comparison ESC 50 dataset

Table 11: The reconstruction performance comparison. We compared with previous SOTA model, such as DAC-Codec and Encodec.

| Model | PESQ | STOI | STFT loss |
|---|---|---|---|
| DAC (3 RVQ) | 1.88 | 0.52 | 1.55 |
| Encodec (3 RVQ) | 2.01 | 0.54 | 1.39 |
| ours | 2.09 | 0.52 | 1.38 |

# F   Potential Negative Societal Impacts

This paper aims to build a multi-modal LLM for audio understanding and generation tasks in a few-shot style without any parameter update for LLMs, which will ease the effort to develop different special models for different tasks. A negative impact is the risk of misinformation. To alleviate it, we can train an additional classifier to discriminate the fakes. We believe the benefits outweigh the downsides. The proposed method lowers the requirements for designing many special models, which may cause unemployment for people with related occupations, such as audio engineers.

