# OpenReview forum: "UniAudio 1.5: Large Language Model-Driven Audio Codec is A Few-Shot Audio Task Learner"
_NeurIPS.cc/2024/Conference — NeurIPS 2024 poster_

### Official Review · Reviewer_XMxu · 2024-07-08

**Soundness:** 2
**Presentation:** 3
**Contribution:** 2
**Rating:** 6
**Confidence:** 3

**Summary:**

The paper proposes a LLM-codec module that can plug into an existing LLM, i.e., LLAMA-2, to perform few-shot in-context learning for tasks including classification (emotion & sound event), and text-to-speech synthesis. The proposed module takes the raw audio waveform as an input, and encodes it into latent space where the corresponding features are mapped into VQ codebooks in the LLM dictionary space. A multi-layer alignment designs are considered in the LLM-codec, where the shallow layer is responsible for semantic information, and deeper layers are responsible for more fine-grained information. Four losses are consider in aligning the features with the LLM pretrained embedding space, semantic loss, consistency loss, reconstruction loss and discriminator loss. For semantic loss and consistency loss, the goal is to guide the learned embeddings to have semantic meaning as well as align with the pretrained audio features for stability reason. Experiments demonstrate that the proposed module can be plugged into the pretrained LLAMA-2 for in-context few-shot learning for simple audio understanding tasks and TTS task.

**Strengths:**

1. The paper proposes an interesting way of solving few-shot audio-related tasks using frozen LLMs. Different from previous methods, this work designs a plug-in module for LLM in-context learning for audio modality to avoid LLM training or fine-tuning. The plug-in module is designed to be efficient, with only 160M parameters.

2. The tasks can cover both audio understanding and simple text-to-speech synthesis, which is flexible enough considering the limitations of in-context learning.

3. The paper is mostly well-written and presentation is clear enough to understand the motivation and the proposed method.

**Weaknesses:**

1. The task seems to be really simple, I am wondering how could the model performs under the more challenging scenarios as N-way goes larger? Also how the TTS performance degrades as the scripts become more complicated?

2. In table 2, for 2-way speech emotion classification, why is random guess only 40% accuracy rather than a number close to 50%. 59% acc is also not high enough for binary classification task. What is missing here in order to achieve better acc?

3. In Table 4, what is ACC for GT and FastSpeech 2, is the number not be able to compute?

**Questions:**

See weaknesses above.

---

> ### Author Rebuttal · Authors · 2024-08-07
>
> We thank the reviewer for recognizing our contributions. We appreciate the constructive comments the reviewer provided to us to improve our paper further. We are delighted to have the following discussion with the reviewer.
>
> **Q1:**  The task seems to be really simple, I am wondering how could the model performs under the more challenging scenarios as N-way goes larger?
>
> **A:**  Thank you for your comments. We set different settings $N=2,3,4,5,6$ for the sound event classification task. The results are as follows. We can see that in the more complex scenarios, our proposed model can also get better performance than the baseline BLSP [1]. Furthermore, inspired by reviewer Gqom, we also find that using a larger LLM (e.g. LLAMA 2 13B) can further improve the performance.
> |   Model / task   | 2-way-1-shot  | 3-way-1-shot | 4-way-1shot | 5-way-1-shot | 6-way-1-shot |
> |:----------------:|:-------------:|:------------:|:-----------:|:------------:|:------------:|
> |  Ours (LLAMA 7B) |       60      |      41      |      36     |      33      |      17      |
> | Ours (LLAMA 13B) |       **62**      |     **42**     |      **41**     |      **43**      |      **31**      |
> |       BLSP       |       47      |      26      |      15     |      12      |      10      |
>
> **Q2:**  Also how the TTS performance degrades as the scripts become more complicated?
>
> **A:**  Thank you for your comment. Inspired by your suggestion, we set three difficulty levels to test the simple text-to-speech. The first level includes addition, subtraction, multiplication, and division, such as the speech of *2x2*. The second level includes simple reasoning, such as *what is the next for the sequence 0,1,2,3*. The third level includes complex reasoning, such as *if 2 times a number plus 1 equals 1, what is the number*. We ask ChatGPT to help construct the questions for each level. For each level, we test 20 utterances. As a result, we find that the accuracy for each level is 85\%, 65\%, and 60\%.
>
> **Q3:**  In table 2, for 2-way speech emotion classification, why is random guess only 40 \% accuracy rather than a number close to 50 \%.
>
> **A:**  Thank you for your comment. As we introduced in the Table 2 caption, for the Random guess, we follow the previous work Dynamic-SUPERB [2], and we calculate the average score based 5 times evaluation. We agree with the reviewer's view that when we run the random guess enough times, the final accuracy will close to 50\%. In order to give the reader a better understanding, we will update the random guess score as the theoretical probability.
>
> **Q4:**  59\% acc is also not high enough for binary classification task. What is missing here in order to achieve better acc?
>
> **A:**  Thank you for your comment. We have two reasons: 1) out-of-domain data leads to lower performance; 2) low understandablity of LLM to audio modality. We have following potential solutions to address these two issues: 1) providing high-diversity audio dataset for model training; 2) fine-tuning the LLM.
>
> **Q5:**  In Table 4, what is ACC for GT and FastSpeech 2, is the number not be able to compute?.
>
> **A:**  For FastSpeech 2, we directly input the ground truth answer text to the TTS model, so the generated speech is always right. When comparing with FastSpeech 2, we want to show that our proposed method can generate high-quality speech.
>
> [1] Wang C, Liao M, Huang Z, et al. Blsp: Bootstrapping language-speech pre-training via behavior alignment of continuation writing[J]. arXiv preprint arXiv:2309.00916, 2023.
>
> [2] Huang C, Lu K H, Wang S H, et al. Dynamic-superb: Towards a dynamic, collaborative, and comprehensive instruction-tuning benchmark for speech[C]//ICASSP 2024-2024 IEEE International Conference on Acoustics, Speech and Signal Processing (ICASSP). IEEE, 2024: 12136-12140.

---

> > ### Comment · Reviewer_XMxu · 2024-08-08
> >
> > I have read through the authors’ response and it addresses all of my questions clearly by providing sufficient further experiment results. I keep my original rating and would lean towards the acceptance of the paper due to its novelty of design in LLM-Codec, albeit needing some improvement in presentation.

---

> > > ### Author Response · Authors · 2024-08-08
> > >
> > > We thank the reviewers for the time and effort. We will further improve our presentation in the final version.

---

### Official Review · Reviewer_4YYB · 2024-07-12

**Soundness:** 1
**Presentation:** 1
**Contribution:** 3
**Rating:** 5
**Confidence:** 4

**Summary:**

The paper introduces LLM-Codec, which enables frozen LLMs to perform various audio tasks in a few-shot manner without fine-tuning LLMs. LLM-Codec operates in a RVQ-manner, hierarchically converting audio tokens into words or sub-words in the LLM vocabulary to compress the audio modality into the text space.

**Strengths:**

The approach is validated through experiments on tasks such as speech emotion classification, audio classification, text-to-speech generation, and speech enhancement, demonstrating its feasibility and effectiveness.

**Weaknesses:**

It’s hard to understand the meaning of the model setup. The authors train the system with distillation from T5 and Whisper instead of mapping them to the text-LLM like BLIP does. What is the motivation for building a system in this pipeline? Why is the decoder needed? What is the advantage of this approach compared to using external TTS or speech enhancement modules? These questions are difficult to answer with the current version of the presentation and authors should compare with BLIP-like approach to show its benefits.

From a presentation perspective, there are many grammatical errors and misleading sentences throughout the manuscript. The authors should put more effort into clarifying their claims and making the manuscript easier to comprehend by providing solid details.
- The captions of Figures 1 and 2 are hard to understand and contain grammatical errors.
- The details of the experimental setup are also insufficient to fully understand the setting.
- The purpose of Figure 4 is very unclear. There is no apparent correlation between the outputs of the semantic layer and the given audio. The purpose of this analysis is not evident, and the information provided is incomplete.

The authors conducted various downstream tasks, yet, the setup of experiments are insufficient to validate the system’s capacity. I’ve pointed out questionable setups at the Questions section.

**Questions:**

- Why is RVQ setting adopted as the system pipeline? Have the authors try comparing with just performing downstream tasks using the outputs from T5 and Whisper?
- for Table 1,
    - What is the evaluation dataset being used? Does it only contain speech data?
    - How about performance comparisons with other metrics such as me reconstruction loss or SI-SDR?
    - How about using better configurations of the baseline models? (e.g., 44K DAC)
    - I didn’t fully get what 3 Vanilla RVQ means for Encodec_24k and DAC_16k.
- for Table 3, why is accuracy the only metric? How about other metrics such as AUC?
- for Table 4,
    - How is ACC being computed? How about WER?
    - What is the intuition of proposed method having better DNSMOS than the GT? Why not compute on subjective MOS?

**Limitations:**

Authors included their potential limitations in the Appendix section.

---

> ### Author Rebuttal · Authors · 2024-08-07
>
> We greatly appreciate the reviewer's time and patience with our paper. We do find these suggestions constructive and helpful.
>
> **Q1:** It’s hard to understand the meaning of the model setup ... What is the motivation for building a system in this pipeline?
>
> **A:**  We apologize that our presentation cannot make the reviewer better understand our model setup and design. We are happy to discuss this with the reviewer. From the high-level, we aim to turn the powerful LLM into a universal audio understanding and generation model to tackle infinite audio tasks without training. Due to the length limitation,  we give the detailed explanation on global response part.
>
> **Q2:**  Why is the decoder needed?
>
> **A:**  Thanks for this comments. As we discussed, we expect the LLMs can generate audio directly. So we first use our LLM-Codec to quantize the audio data into the LLM's token space (word or sub-word). Then LLMs will predict the corresponding tokens based on the instruction, these tokens can be recovered into waveform by using the codec decoder. Such a strategy has been widely used in audio generation, such as AudioLM [5], researchers first use a codec model to compress the audio into discrete tokens, then use a transformer model to predict the corresponding tokens. Lastly, the codec decoder is used to recover waveform from the predicted tokens.
>
> **Q3:**  From a presentation perspective, there are many grammatical errors and misleading sentences throughout the manuscript. ....
>
> **A:**  We appreciate this comment. We are glad to revise our paper for better writing clarity.
>
> **Q4:** Why is RVQ setting adopted as the system pipeline?
>
> **A:**  Thank you for this comment. The reason includes (1) we need to quantize audio into discrete tokens (2) RVQ is the mainstream approach for audio quantization. Furthermore, we use the LLM's vocabulary as the RVQ codebook, which means that we quantize the audio modality into the LLM's vocabulary. The benefits include: (1) with the help of LLM-Codec, LLMs can directly generate audio; (2) we do not need to fine-tune the LLMs.
>
> **Q5:**  Have the authors try comparing with just performing downstream tasks using the outputs from T5 and Whisper?
>
> **A:**  Yes, one of our baseline BLSP [1], which using a Whisper encoder to extract speech representation, and fine-tuning the LLM with a learnable adaptor. Due to BLSP only output text, we only compare with it on audio understanding tasks, such as speech emotion classification and sound event classification.
>
> **Q6:**  What is the evaluation dataset being used? Does it only contain speech data?
>
> **A:**  We appreciate this comment. As we introduced in Table 1 caption, we evaluate the reconstruction performance on the VCTK dataset. We randomly chose 200 utterances. Inspired by your suggestion, we also chose 200 utterances from the ESC50 dataset to evaluate the reconstruction performance on sound data. The results as Table 2 shows (refer to global response PDF).
>
> **Q7:**  How about performance comparisons with other metrics such as mel reconstruction loss or SI-SDR?
>
> **A:**  Based on your suggestion, we add the Mel reconstruction loss metric as one of the evaluation metrics. The performance as Table 3 shows. We will update this into our final version.
>
> **Q8:** How about using better configurations of the baseline models? (e.g., 44K DAC)
>
> **A:**  We agree that the 44K DAC model has good reconstruction performance for high-sampling rate audio (e.g. 44.1k hz). However, in our study, we train all of the codec models on 16khz audio data. Thus, we choose 16K DAC-codec as one of the baselines. Due to the 16K Encodec model is not released, we downsampling the generated audio by Encodec_24k into 16k.
>
> **Q9:**  I didn’t fully get what 3 Vanilla RVQ means for Encodec_24k and DAC_16k.
>
> **A:** We thank this important comment and appreciate the reviewer. In our study, we propose a multi-scale residual vector quantization, which is different from previous commonly used RVQ in DAC and Encodec, so we name previous RVQ as Vanilla RVQ. We apologize for this misunderstanding, we will add an explanation to the Table 1 caption part.
>
> **Q10:**  for Table 3, why is accuracy the only metric? How about other metrics such as AUC?
>
> **A:**  For the metric, we follow the baseline Dynamic-superb to use accuracy as the metric. We agree that AUC is a good metric to evaluate a classifier. We want to point out that AUC is calculated by setting different threshold values to get a group of FPR and TPR. However, we use the LLMs to directly predict the text label,  it is hard to set a 'threshold' like a traditional classification model. In other words, AUC is more suitable to evaluate the performance of traditional classifiers, in which humans can decide the threshold. If the reviewer can provide better metrics, we are happy to add them to the paper.
>
> **Q11:** for Table 4, How is ACC being computed? How about WER?
>
> **A:**  We calculate the ACC by comparing the content of the generated speech with the ground truth value. Actually, due to the generated speech only includes digits, using ACC and WER has the same meaning.
>
> **Q12:** What is the intuition of proposed method having better DNSMOS than the GT? Why not compute on subjective MOS?
>
> **A:**  The reason is that the GT from the Free Spoken Digit Dataset (FSDD) is low-quality. FSDD is an old speech digit dataset, which may include some noise. Instead, our audio codec model is trained on a clean and high-quality speech dataset, so the noise details will not be modeled by the codec model. Actually, such a phenomenon widely exists in speech generation, nowadays, many TTS models can synthesize better speech than the original one, e.g. NaturalSpeech 3 [8] shows it can generate better speech than LibriSpeech. Inspired by your valuable suggestion, we conducted a subjective evaluation as Table 4 shows.
>
> Due to the length limitation, we put the remaining question into global response part.

---

> > ### Comment · Reviewer_4YYB · 2024-08-12
> >
> > I appreciate the authors' efforts in their rebuttal and have carefully read through all the reviews and additional clarifications provided. This has given me a clearer understanding of the paper's contributions.
> >
> > However, my primary concern remains with the paper's presentation. The current version lacks clarity, which makes it difficult to fully appreciate the work's contributions. For this reason, I remain hesitant to recommend a solid acceptance. It is crucial that the final version significantly improves the clarity and presentation to effectively communicate the findings.
> >
> > That said, I do acknowledge the interesting empirical findings presented within the existing pipeline, which has led me to slightly raise my score.

---

> > > ### Author Response · Authors · 2024-08-13
> > >
> > > Dear Reviewer 4YYB,
> > >
> > > Thank you again for your tremendous efforts and valuable comments. We sincerely appreciate your recognition of our contributions and your constructive feedback. We are committed to improving our presentation based on your comments and suggestions.
> > >
> > > We are currently revising the manuscript according to your feedback.  We hope this will address your concerns. We will continue refining our presentation in the coming days to ensure a polished final version.
> > >
> > > In this version, we have mainly updated the following sections:
> > >
> > > **1. We carefully reviewed our abstract and introduction to ensure proper grammar and enhance readability.**
> > >
> > > **2. We updated the captions of Figures 1, 2, and 4. We hope the revised captions provide better clarity and address your concerns.** The details as follows:
> > >      **Figure1:** This figure illustrates the framework of the proposed approach for performing speech emotion classification and simple text-to-speech generation tasks. For each task, we prepare the instruction, demonstrations (e.g., x_1, y_1, x_2, y_2 ), and the query x_q. The LLAMA 2 model is then asked to predict the corresponding result y_q. Here, y_q can be either text or audio.
> > >
> > >   **Figure 2:** This figure provides a high-level overview of LLM-Codec, including an encoder, a decoder, a multi-scale discriminator, and a multi-scale residual VQ layers. Here, ‘sub’ denotes feature subtraction. Note that the modules marked with a snowflake are frozen during training.
> > >
> > > **Figure 4:**  The token visualization of the semantic layer of LLM-Codec is shown. We present two groups of samples, each containing two audio recordings with the same sound event label. In each group, we use the same color to highlight potentially similar patterns in the two audio recordings, such as identical token sub-sequences or token repeating frequencies. We speculate that these patterns can be easily recognized by LLMs, allowing them to learn new sound events quickly with just a few demonstrations.
> > >
> > > **We rewrote the Experimental Setting section to improve clarity. This section is now divided into two subsections: The first subsection provides detailed information about LLM-Codec, including the training data, codec model configuration, evaluation data, evaluation metrics, and the corresponding audio codec model baselines. The second subsection details the integration of LLM-Codec with pre-trained LLMs for downstream tasks (e.g., emotion classification, sound event classification, text-to-speech, etc.), including the evaluation data for each downstream task and the compared baselines..**
> > >
> > > Once again, we greatly appreciate that you raised the score and believe your valuable comments have significantly improved the paper, offering more precise explanations and presentations. We sincerely thank you for your time, effort, and patience during this peer review process.  We are always happy to have a further discussion and answer more questions raised by you.

---

### Official Review · Reviewer_oXws · 2024-07-13

**Soundness:** 3
**Presentation:** 4
**Contribution:** 3
**Rating:** 6
**Confidence:** 2

**Summary:**

The paper introduces LLM-Codec, a novel audio codec model that leverages Large Language Models (LLMs) to perform various audio tasks with minimal training examples.

By translating audio signals into the token space of LLMs, it enables these models to understand and generate audio content.

The model uses a multi-scale residual vector quantization approach to maintain audio quality while reducing token sequence length.

**Strengths:**

- This paper shows a novel codec model for audio compression and can be generated by a frozen LLM for in-context learning.

- The task setting is challenging, and the semantic codec is a challenging task, while the paper shows good exploration to it.

- The ablation study is sufficient, showing the elements in the RVQ codec model's usages.

**Weaknesses:**

Experiments.

Considering the paper proposes an encodec model, the most important result is the reconstruction performance. Providing Table 1, the necessary explaination of the results is lacking. I suggest this paper adds additional demonstration to the experimental main results to improve the readability.

Although the paper does a lot of experiments, showing promising results, the audio generation-related tasks still lack enough experiment results. In both introduction and related work section, the paper claims that previous codec models do not support  audio generation tasks, while the text-to-audio evaluation results are not shown. Considering the strong claim and the AudioCaps training data, it is strongly recommended to show its performance comapred to other text-to-audio models.

**Questions:**

N/A

**Limitations:**

This paper provides sufficient discussion in this field.

---

> ### Author Rebuttal · Authors · 2024-08-07
>
> We thank the reviewer for recognizing our contributions. We appreciate the constructive comments the reviewer provided to us to improve our paper. We are delighted to have the following discussion with the reviewer.
> **Q1:**  Considering the paper proposes an encodec model, the most important result is the reconstruction performance. Providing Table 1, the necessary explaination of the results is lacking. I suggest this paper adds additional demonstration to the experimental main results to improve the readability.
>
> **A:**   We thank the reviewer for his/her valuable suggestions. We are glad to revise our paper to improve its readability. Specifically, we would:
>
> *(1)* Provide more explanation for reconstruction performance results in Section 4.2, including the reconstruction performance comparison, the influence of down-sampling steps and tokens per second. Furthermore, inspired by reviewer 4YYB, we add a new metric (the Mel reconstruction loss) into Table 1.
>
> *(2)* Highlight the advantages of our proposed codec.
>
> **Q2:** Considering the strong claim and the AudioCaps training data, it is strongly recommended to show its performance comapred to other text-to-audio models.
>
> **A:**   We appreciate this suggestion. We add a text-to-audio evaluation. Specifically, we choose previous SOTA AudioGen [1] as one of the baselines, because AudioGen is also an autoregression model based on audio codec models. Furthermore, we also choose some diffusion-based audio generation models, including AudioLDM [2] and Tango [3], as the other baselines. For AudioLDM and Tango, we use their official checkpoints, and we set 200 diffusion steps for the inference. We conduct experiments on the ESC 50 [4] validation set. AudioGen, AudioLDM, Tango, and our model do not see the ESC 50 dataset in the training stage. We use the event label to construct the text description, e.g. if the event label is 'clapping', we will construct the caption as 'this is the sound of clapping.'. For the evaluation metrics, we follow previous works to use FAD and KL as the metrics. The results are shown in the following table. Furthermore, we also add a visualization of generated samples in Figure 1. We will update this into our final version.
>
> |   model  |  FAD  |  KL  |
> |:--------:|:-----:|:----:|
> | AudioGen |  20.4 | **1.94** |
> | AudioLDM |  15.6 | 3.52 |
> |   Tango  |  **12.7** | 3.01 |
> |   ours   | 17.16 | 3.05 |
>
> [1] Kreuk F, Synnaeve G, Polyak A, et al. Audiogen: Textually guided audio generation[J]. ICLR 2023.
>
> [2] Liu H, Chen Z, Yuan Y, et al. Audioldm: Text-to-audio generation with latent diffusion models[J]. ICML, 2023.
>
> [3] Ghosal D, Majumder N, Mehrish A, et al. Text-to-audio generation using instruction-tuned llm and latent diffusion model[J]. ACM-MM, 2023.
>
> [4] Piczak K J. ESC: Dataset for environmental sound classification[C]//Proceedings of the 23rd ACM international conference on Multimedia. 2015: 1015-1018.

---

> > ### Comment · Reviewer_oXws · 2024-08-13
> >
> > Thanks for your explanation. I appreciate your efforts to add experiments comparing with the existing audio generation models. It is because that only when the related experiments are conducted, we can say if the paper overclaims its generation ability, which is states in the introduction section.
> >
> > From the results, we can find that the current model's performance is worse than a common baseline, AudioLDM 1. Considering the results, I personally suggest that the paper may use the term "support audio generation tasks" more carefully.
> >
> > Hence, I will not improve or reduce the current score. Overall it is a very good paper discussing a useful topic.

---

> > > ### Author Response · Authors · 2024-08-13
> > >
> > > Dear Reviewer oXws,
> > >
> > > Thank you again for your tremendous efforts and valuable comments. We sincerely appreciate your recognition of our contributions and your constructive feedback.
> > >
> > > We agree that the current model’s performance in the text-to-audio task is still below that of previous specialized models, such as AudioGen and AudioLDM. We believe one reason for this is the difference in data coverage. In our study, we utilized only the AudioCaps dataset, whereas other specialized models have leveraged more extensive data sources, such as AudioSet. Therefore, a potential direction for improvement would be to scale up the data coverage.
> > >
> > > Once again, we greatly appreciate that you recognize our contributions.  We sincerely thank you for your time, effort, and patience during this peer review process. We are always happy to have a further discussion and answer more questions raised by you.

---

### Official Review · Reviewer_Gqom · 2024-07-15

**Soundness:** 4
**Presentation:** 3
**Contribution:** 3
**Rating:** 7
**Confidence:** 5

**Summary:**

The authors introduce a three-step audio-to-discrete codec to encode continuous acoustic information into a form suitable for large language model-based audio and speech understanding.

Overall, this method is novel and represents an important step in audio modeling. The architecture targets different levels of acoustic information, from semantics to acoustic representation, using only discrete codecs. The results are empirically strong and solid.

However, there are fewer theoretical connections to justify the meaning of the lexical representation of the "trainable new (pseudo) language" of speech, which can be improved in future work.

For example, former works on word-level model reprogramming, learning equivalent pseudo tokens [A] from random embeddings, and theoretical bounds [B] on connecting different layers for latent alignment (e.g., when and how to align these three RVQ adapters) could help justify the uniqueness of the latent distance compared to embedding injection-based methods. This analysis could strengthen the theoretical foundations of the paper.

Despite these points, the overall contributions remain high-quality.

- A few grammatical and formatting issues need to be very fixed in the final draft for a much ready version.

In sum, I highly recommend accepting this paper and suggest the authors address these issues in the final version.

**Strengths:**

1. the model architecture design is overall new and effective

2. the designs on the different concept of audio representation is interesting

**Weaknesses:**

1. there are less discussion on the representation difference of proposed codec-based method to the embedding injection based works.

2. some minor grammar and formation issues

3. there are less theoretical discussion on the representation related to how many RVQ layers needed eventually

**Questions:**

See the weakness (to be improved)

1. Are there any streaming or token merging limitation?

2. Is there any scaling effects of the backbone LM selection?

3. what would be the semantic alignment from the codec to the lexical level representations?

**Limitations:**

minor, in terms of performance, the gap between cascaded LM for ASR and translation tasks.

---

> ### Author Rebuttal · Authors · 2024-08-07
>
> We thank the reviewer for recognizing our contributions. We appreciate the constructive comments the reviewer provided to us to improve our paper further. We are delighted to have the following discussion with the reviewer.
> **Q1:**  A few grammatical and formatting issues need to be very fixed in the final draft for a much-ready version.
> **A:**  We appreciate this comment. We are glad to revise our paper for better writing clarity.
>
> **Q2:** there are fewer theoretical connections to justify the meaning of the lexical representation of the "trainable new (pseudo) language" of speech, which can be improved in future work. For example, former works on word-level model reprogramming, learning equivalent pseudo tokens [1] from random embeddings and theoretical bounds [2] ...
> **A:**  We appreciate and agree with the reviewer's suggestion. We also want to build a theoretical foundation for this study. We plan to refer to your mentioned reference paper pseudo tokens [1] and theoretical bounds [2], and try to learn how to build a theoretical foundation in the future. Can you give a detailed paper name or URL for [1] and [2]? We very much appreciate your help.
>
> **Q3:**   There is less discussion on the representation difference between the proposed codec-based method and the embedding injection-based works.
> **A:**  We appreciate and agree with the reviewer's comment: we will add a discussion section to show the difference between our proposed codec-based method and previous embedding injection-based works, e.g. previous works (e.g. BLSP [1] or Qwen-Audio [2]) use Whisper model extracts continuous embedding, and then fine-tuning pre-trained LLM on audio understanding tasks. We will add the following content to show the difference:
>
> *(1) Motivation:* the embedding injection-based works expect that LLMs can understand audio embedding by fine-tuning LLMs so that their model supports input audio modality and output text content. Instead, our proposed method tries to transfer audio modality into LLM's token space, so that LLM can directly understand audio modality without additional fine-tuning.
>
> *(2) Formulation:* the embedding injection-based works need a fine-tuning LLMs stage. In general, they need to train an adaptor or fine-turn part of the parameter by the LORA strategy. Instead, our proposed method does not need any parameter updating for LLMs.
>
> *(3) Target:* most of the embedding injection-based works focus on audio understanding tasks. Instead, we expect to build a universal audio understanding and generation framework with the help of LLM-Codec.
>
> **Q4:**  there are less theoretical discussion on the representation related to how many RVQ layers needed eventually
> **A:**  We appreciate and agree with the reviewer's comment. We are happy to discuss this issue: From a reconstruction performance perspective, using more RVQ layers will bring better performance. From a generation perspective, using more RVQ layers brings the long sequence problem for LM, furthermore, it also increases the inference costs. Thus, we seek a compact but complete audio representation, preserving sufficient semantic and acoustic information with few tokens.
>
> **Q5:**  Are there any streaming or token merging limitation?
> **A:**   We are happy to discuss this issue with the reviewer. Previous codec models, such as Encodec and SoundStream, are both supporting streaming. One of the reasons is that these works adopt a causal convolution block in the Encoder and Decoder parts. In our codec, we also use similar convolution blocks with Encodec. So our codec also supports streaming. For the token merging limitation, we adopt the multi-scale RVQ strategy, which results in different VQ layers producing different numbers of tokens, which may bring challenges in token merging.
>
> **Q6:**   Is there any scaling effects of the backbone LM selection?
> **A:**    We appreciate the reviewer's comment. Inspired by your suggestion, we added an experiment to explore the influence of scaling effects of the backbone LM. Specifically, we compare the performance of different LM selections: LLAMA2 7B and LLAMA 2 13B. We conduct experiments on N-way-1-shot sound event classification, The performance comparison as following Table shows. We can see that scaling the backbone LM can also bring improvement for audio tasks.
> |   Model / task   | 2-way-1-shot  | 3-way-1-shot | 4-way-1shot | 5-way-1-shot | 6-way-1-shot |
> |:----------------:|:-------------:|:------------:|:-----------:|:------------:|:------------:|
> |  Ours (LLAMA 7B) |       60      |      41      |      36     |      33      |      17      |
> | Ours (LLAMA 13B) |       **62**      |     **42**     |      **41**     |      **43**      |      **31**      |
>
> **Q7:**  what would be the semantic alignment from the codec to the lexical level representations?
> **A:**    We appreciate the reviewer's comment. In our view, the semantic token should have a strong connection or high correlation with the lexical-level representations. For instance, the same semantic information in two audios should be mapped into a similar lexical sequence, so that the LLMs can learn the the pattern with few demonstrations.
>
> [1] Wang C, Liao M, Huang Z, et al. Blsp: Bootstrapping language-speech pre-training via behavior alignment of continuation writing[J]. arXiv preprint arXiv:2309.00916, 2023.
>
> [2] Chu Y, Xu J, Zhou X, et al. Qwen-audio: Advancing universal audio understanding via unified large-scale audio-language models[J]. arXiv preprint arXiv:2311.07919, 2023.

---

> > ### Comment · Reviewer_Gqom · 2024-08-11
> >
> > Thanks for the authors’ response. I think the original suggested references are missing due to some open review formatting.
> >
> > On the token level exploration, the most representative work is WRAP in ACL 2021 and the first speech model prompting work, Voice2Series in ICML 2021 has provided a general population risk bound for 1d vector discrete matching via measurement.
> >
> > The authors could strength their work with wider audiences based on more in-depth connections on these two well known works.
> >
> > But I think the current version is acceptable for my evaluation for Neurips; although with relatively shallow theoretical findings.
> >
> > I recommend to accept this work and please add these extra discussion in a final version.

---

> > > ### Author Response · Authors · 2024-08-11
> > >
> > > Dear Reviewer Gqom,
> > >
> > > Thank you again for your great efforts and the valuable comments. Your suggestion significantly improves our work. We will add the extra discussion in the final version.

---

### Author Rebuttal · Authors · 2024-08-07

We thank the meta-reviewer for organizing this helpful peer review stage. We thank all reviewers for their time, patience, and constructive comments to help us improve our paper.

We specifically address the concerns raised by reviewers **4YYB** regarding the motivation of our paper. We are eager to engage in the following discussions:

**What is the motivation for building a system in this pipeline?**

As we discussed in the Introduction part. The success of LLMs inspires many researchers to build multi-modal LLMs to solve audio-related tasks in the audio domain. For instance, BLSP [1], and Qwen-Audio [2] typically use a pre-trained Whisper encoder to extract speech embedding and then use a learnable adaptor module (e.g. Linear layer or Q-former [3]) and LORA to map the speech embedding into the representation space of LLM. Such a strategy has been widely discussed and recognized in the research community. However, previous works (1) focus more on expanding LLMs to solve specific audio tasks, without considering the in-context-learning ability to unseen audio tasks, e.g. Quen-Audio collects over 30 tasks to conduct multi-task training. Their motivation is using the strong ability of LLMs to help improve the performance of audio understanding tasks. (2) do not support audio generation tasks, which limits its application scenarios. In general, audio tasks can be divided into audio understanding and audio generation. We refer to the audio understanding task as the model's input can be text and audio, but the model's output is only text, e.g. Automatic speech recognition (ASR), Spoken language identification, Emotion recognition, and so on. Similarly, if the model's output is audio, we call these tasks as audio generation tasks, e.g. text-to-speech (TTS), text-to-sound, speech enhancement, and so on. We claim that  **one of our motivations is building a universal audio understanding and generation tasks solver with the help of LLMs and the proposed LLM-Codec**. We highlight that making LLMs to generate audio is not an easy thing. Some pioneer works, such as SpeechGPT [4], try to make a pre-trained LLAMA2 model generate speech. To realize this target, SpeechGPT expands its vocabulary with speech tokens and uses large-scale datasets and GPUs to learn the alignment between speech and text. In contrast, we propose to train an audio codec that can quantize the audio modality into LLM's token space, so that we do not need to expand the LLM's vocabulary like SpeechGPT.  **In summary, we aim to turn the powerful LLM into a universal audio understanding and generation model to tackle infinite audio tasks without training.** To realize this target, we propose to map the audio modality into LLM's token space. As a result, we only use 2 GPUs to train the codec model, and one GPU to conduct the inference with pre-trained LLMs.

**What is the advantage of this approach compared to using external TTS or speech enhancement modules?**

We thank this important comment and appreciate the reviewer. We agree that using external TTS or speech enhancement modules can also build a cascade system. However, our motivation is to build a universal end-to-end system, which supports audio and text as input and output. We expect our model can handle multiple tasks. We understand and agree with the reviewer's opinion: combining multiple external modules can also solve many tasks, such as HuggingGPT [6]. But we want to reach a consensus with the reviewer: combining multiple external modules or building a universal model (such as GPT4-o) are both potential paths toward artificial general intelligence (AGI). As a research work, we expect to explore more potential possibilities and inspire more work.

**Furthermore, we also summarize all of the response Table and Figure as one PDF file**

[1] Wang C, Liao M, Huang Z, et al. Blsp: Bootstrapping language-speech pre-training via behavior alignment of continuation writing[J]. arXiv preprint arXiv:2309.00916, 2023.

[2] Chu Y, Xu J, Zhou X, et al. Qwen-audio: Advancing universal audio understanding via unified large-scale audio-language models[J]. arXiv preprint arXiv:2311.07919, 2023.

[3] Li J, Li D, Xiong C, et al. Blip: Bootstrapping language-image pre-training for unified vision-language understanding and generation[C]//International conference on machine learning. PMLR, 2022: 12888-12900.

[4] Zhang D, Li S, Zhang X, et al. Speechgpt: Empowering large language models with intrinsic cross-modal conversational abilities[J]. EMNLP, 2023.

[5] Borsos Z, Marinier R, Vincent D, et al. Audiolm: a language modeling approach to audio generation[J]. IEEE/ACM transactions on audio, speech, and language processing, 2023, 31: 2523-2533.

[6] Shen Y, Song K, Tan X, et al. Hugginggpt: Solving ai tasks with chatgpt and its friends in hugging face[J]. Advances in Neural Information Processing Systems, 2024, 36.

[7] Huang C, Lu K H, Wang S H, et al. Dynamic-superb: Towards a dynamic, collaborative, and comprehensive instruction-tuning benchmark for speech[C]//ICASSP 2024-2024 IEEE International Conference on Acoustics, Speech and Signal Processing (ICASSP). IEEE, 2024: 12136-12140.

[8] Ju Z, Wang Y, Shen K, et al. Naturalspeech 3: Zero-shot speech synthesis with factorized codec and diffusion models[J]. ICML, 2024.

---

### Decision · Program_Chairs · 2024-09-25

**Decision:**

Accept (poster)

**Comment:**

The paper presents LLM-Codec, an innovative audio codec model that utilizes Large Language Models (LLMs) to handle various audio tasks with minimal training data. By converting audio signals into the token space of LLMs, the model enables these language models to comprehend and generate audio content.

Four reviewers assessed this work, and there was a good discussion period. The authors have sorted out several key concerns raised by the independent reviewers, and some reviewers have modified their overall evaluation of the paper. Unfortunately, two key issues still remain unsolved, and both are acknowledge by the authors during the discussion phase, namely: (i) poor presentation -  it needs improvement, and (ii) current model's performance is worse than a common baseline, AudioLDM -  the authors' claim should be then tuned down. Nonetheless, all reviewers agree that the topic is interesting and the idea can be considered novel.